# Energy Use and Its Key Factors in Hotel Chains

**Rodrigo Schons Arenhart [1,]***[ID]**, Adriano Mendonça Souza [2] and Roselaine Ruviaro Zanini [2]**

[1] Department of Production Engineering and Systems, Federal University of Santa Maria, Roraima Avenue, Santa Maria 1000, Brazil

[2] Statistics Department, Federal University of Santa Maria, Roraima Avenue, Santa Maria 1000, Brazil; amsouza.sm@gmail.com (A.M.S.); roselaine.zanini@ufsm.br (R.R.Z.)

[*] Correspondence: rodrigo.schons@acad.ufsm.br; Tel.:+55-55-99608-1406

**Abstract:** Hotel chains are reported as one of the most energy-intensive sectors and a growing number of international studies on this theme have been developed. This research aims to understand energy use and some of its key factors in hotel chains worldwide. Data were collected on variables related to previous research and those present in the Global Reporting Initiative (GRI) framework. The sample was composed by 45 international hotel chains, representing more than 54,000 properties and 7,500,000 rooms. Multiple linear regression was employed to assess how the predictor variables (water use, carbon intensity, RevPAR, and NetRoom) are associated with energy use (dependent variable). It was presented that hotel chains can pass on the price of energy consumption to their guests, increasing their revenue per available room (RevPAR), but the returns in profitability are not being generated. The RevPAR variable maintained a positive relationship, +0.244, with energy use in the first regression model, with $R^2$ adjusted equal to 0.9506, while the net profit per room (NetRoom) presented a negative relationship in both models, $-0.0006$ and $-0.0010$, respectively, with $R^2$ adjusted equal to 0.9304 in the second model. Investing in updating their energy systems, hotel chains can contribute to a more sustainable future, build positive marketing, retain guests, and generate a long-run financial return. This research contributes to the scientific literature by confirming relationships and providing evidence among new, and not yet explored, variables. It is expected to create a reference for policies to reduce energy use in hotels and for hotel owners to upgrade their systems.

**Keywords:** hotel chains; energy use; Global Reporting Initiative; sustainability; statistical analysis

## 1. Introduction

The hospitality industry is a worldwide source of income and a larger energy user. The tourism industry was responsible for creating 330 million jobs worldwide and represented 10.3% of the world's gross domestic product in 2019 [1]. Despite this industry being made up of small operations, collectively, the resource consumption and the environmental impact are significant [2]. The growth in energy use has resulted in several environmental problems, including an increase in greenhouse gas emissions (GHG), which is a major factor in global warming [3]. Some energy systems are not optimized and thus contribute to a high carbon footprint, which has generated a demand for legislation in energy-use reduction and the use of renewable sources in buildings [4]. Scholars have found that the energy consumption intensity of hotel buildings is the highest and that international tourism has a positive effect on energy use, which should generate a focus on renewable energy production and use [5–7]. Reducing energy use, implementing non-polluting energy systems, and improving efficiency are key to mitigating climate change's effects [8].

An understanding of energy consumption is also crucial for the tourism sector because the high energy cost [9] directly affects accommodation prices and the final profitability of companies. Furthermore, stakeholders call upon firms to assume additional responsibilities for the benefit of the community and environment, generating in the hotel industry a

gradual adoption of corporate social responsibility (CSR) practices [10]. Global carbon reporting frameworks and guidelines were developed to facilitate measurement and reporting processes, such as the Global Reporting Initiative (GRI), which together with a growing number of academic studies, has resulted in a considerable number of companies reporting their climate change performance [11].

Over the last 20 years, many investigations on energy and water use in hotels have been developed; most studies focused on electricity consumption, some analyzed fuel use, and few verified water consumption [12]. Many variables showed interactions with energy use, such as hotels' floor area [3,12–14], guest nights [12,13], star rating [2,15], worker density [2,4,14], occupancy [3,9,12,15], and revenue [4,14,15], among others. Despite this, there is still room for the discovery and ratification of relationships linked to energy consumption in hotels. It was noticed that a major part focused on collecting data from individual hotels, and just one study in the literature is based on data from hotel chains [13]. This occurs due to the lack of data, which can usually be found only in sustainable reports or by conducting surveys. Another characteristic observed in the studies present in the relevant journals was the geographical area, with works that normally cover only one city [15], country [2,4,9,12,14], or region [3,13].

This research intends to provide a better understanding of energy use and its key factors in hotel chains around the world based on multivariate regression models, assessing how water use, carbon intensity, RevPAR, and NetRoom are associated with energy use. The data collection from the sustainable reports of hotel chains was conducted with special attention to the factors presented by previous research. In addition to the variables found in the literature, a new variable to explain energy use is suggested for further research and is included in the model. The results are expected to be one more reference for policymakers' actions to reduce energy consumption in the hospitality industry and to qualify hotel owners for performance checks in their buildings.

## 2. Literature Review

### 2.1. Energy Use in Hotels

A growing number of international articles on resource consumption in the hospitality sector have been published [12]. Several indicators found in research are incorporated in the environmental reports of hotel companies. Despite this, there is a lack of standardization in the data presentation and, in many cases, the omission of important variables. These circumstances make it difficult for stakeholders to understand the complete information and do not help in the development of research.

Relevant studies that analyzed the issue of resource consumption in hotels have been identified in the last 15 years. The main emphasis was given to those works that identified relationships between energy use and other significant variables measured in hotels. These studies collected data in several locations such as Europe, Singapore, Taiwan, Hong Kong, China, Tunisia, and the Canary Islands. All studies developed a methodology to evaluate energy use, with two papers evaluating water consumption as well. The hotel sample varies from 6 to 200 hotels. Table 1 presents prior research on energy use in hotels.

Despite the number of recent studies, only one article used data from hotel chains, which included 184 hotels from two world-renowned brands. Other works used data from surveys in hotel units, without considering the use of energy by hotel conglomerates. Another limitation was the geographical distribution of this research, which ended up reflecting the use of resources in specific regions of the world. To fill this gap, this study aims to provide a better understanding of energy use and its key factors in hotel chains around the world.

Previous research (please see Table 1) has seen some significant relationships between the dependent variable, energy use, and several independent variables. The relationship between energy and the occupancy of the rooms and the revenue generated by them is highlighted, and this positive relationship was seen in five of the eight studies evaluated. In addition, another variable that showed a significant and positive correlation with the depen-

dent variable in four different studies was the floor area. The other variables contributed in a not-so-evident way to the studies.

**Table 1.** Prior research on energy use in hotels.

| Authors | Sample and Period | Region | Dependent Variable(s) | Independent Variable(s) | Significant Result(s) |
|---|---|---|---|---|---|
| [2] | 29 hotels—2004 | Singapore | Energy use | Star rating<br>Worker density<br>Last retrofit | Star rating (−)<br>Worker density (+) |
| [3] | 30 hotels—2013 | Hong Kong | Energy use | Building age<br>Floor area<br>Guestroom<br>Occupancy rate<br>Maintenance costs | Floor area (+) |
| [4] | 6 hotels—2019/2020 | Gran Canaria | Energy use | Overnight stays<br>Number of diners<br>Number of workers<br>Number of rooms<br>RevPAR<br>Pool volume<br>Number of guests per room | RevPAR (+)<br>Number of diners (+)<br>Pool volume (+)<br>Number of guests per room (+) |
| [9] | 73 hotels—2010 | Taiwan | Energy use | Foreign individual travelers (FIT)<br>Number of group guests | FIT (+)<br>Group guests (+) |
| [12] | 55 hotels—2018 | Tunisia | Energy use and water use | Floor area<br>Number of beds<br>Number of guests rooms<br>Number of guests-nights<br>Occupancy rate<br>Floor area (guestrooms) | Floor area (guestrooms) (+)<br>Number of guests-nights (+) |
| [4] | 6 hotels—2019/2020 | Gran Canaria | Energy use | Overnight stays<br>Number of diners<br>Number of workers<br>Number of rooms<br>RevPAR<br>Pool volume<br>Number of guests per room | RevPAR (+)<br>Number of diners (+)<br>Pool volume (+)<br>Number of guests per room (+) |
| [13] | 184 hotels—2004 | Europe | Energy and water use | Floor area<br>Guest-nights sold<br>Food cover sold<br>laundry washed on-site<br>On-site health club | Floor area (+)<br>Guest-nights sold (+)<br>Food cover sold (+) |
| [14] | 200 hotels—2010 | Taiwan | Energy use | Floor area<br>Number of rooms<br>Number of buildings<br>Number of workers<br>Occupancy rate<br>ADR<br>Total revenue<br>Number of guests | Floor area (+)<br>Number of rooms (+)<br>ADR (+)<br>Total revenue (+)<br>Occupancy rate (+) |
| [15] | 24 hotels—2013 | Lijiang, China | Energy use | Floor area<br>Number of guests rooms<br>Star rating<br>Occupancy rate<br>Room revenue<br>Number of workers<br>Floor area (guestrooms) | Star rating (+)<br>Occupancy rate (+) |

## 2.2. GRI Adoption

The GRI guidelines were introduced in 2000 and were revised continuously to provide a standardized framework and to ensure the comparability and consistency of the global reporting [11]. This framework's popularity has grown significantly and can be viewed as the most widely adopted reporting framework currently [16]. As these data are mostly

reviewed by external parties, they contribute to the reliability and applicability of studies related to the financial, social, and environmental areas.

The use of GRI indicators in studies related to the topic was not noticed, except in three works [14]. One reference used the financial indicators ADR (average daily rate), the total revenue, and the occupancy rate [15]. Another reference also used the occupancy rate to predict the energy consumed and reference [4] used the RevPAR (revenue per available room). As none of the prior research consulted made use of GRI metrics other than the financial area, this may be an important investigation factor for current studies. This article uses other environmental indicators extracted from sustainable reports based on GRI, i.e., water use and carbon intensity. Other studies have already verified a relationship between financial indicators and energy use, and this work will use RevPAR and net profit indicators.

Water use was studied as a dependent variable [12,13], but this variable was not used yet as a predictor variable for energy. It is expected to verify if there is a relationship between water consumption and energy use in hotel chains. Another environmental variable that was not seen in previous studies on energy consumption was carbon intensity. This variable is reported in Scope 1, 2, and 3. Scope 1 presents emissions due to stationary and mobile sources of fossil fuels, Scope 2 refers to the use of electricity purchased from utilities and gas use, and Scope 3 refers to lower emission activities [17]. As the GHG is linked to the direct use of hotel facilities and guest accommodation, a significant relationship with energy consumption is expected.

RevPAR is an indicator obtained by dividing total revenue by the number of rooms available, and is a function of occupancy and an average daily rate [18]. For this reason, the use of the ADR and the occupancy rate has no extra relevance to this work. This indicator has already been used before and a positive relationship with energy consumption was verified [4], and with ADR and occupancy rate [14,15]. To investigate the relationship between the profitability of hotel chains and their energy use, a net-profit-per-room indicator was developed. This indicator has not been seen in any previous study and has a potentially unprecedented use in this type of research; it is calculated by dividing the net profit generated by the hotel chain by the number of rooms available. As research with this type of indicator was not found, its potential results cannot be assumed a priori.

## 3. Materials and Methods

### 3.1. Data Collection

The data collection was focused on energy use and different variables potentially impacting it. The independent variables were chosen as previously presented: water use, carbon intensity, RevPAR, and net profit per room (NetRoom). The data were collected from sustainable and annual reports of hotel chains (see Appendix A). A total of 45 hotel chains were selected, with complete information on all indicators surveyed, and the reference year for the information was 2019. The choice of this year was motivated by being the last year before the emergence of the health crisis caused by COVID-19, which impacted severely the hotel industry.

Among these 45 hotel chains, there are more than 54,000 properties and more than 7,500,000 rooms. Thus, despite the sample being considered small for statistical significance, it is perceived that it has great practical relevance, as it represents the largest hotel companies and a considerable part of worldwide hotels. Even with the adoption of the GRI framework, there was a lack of standardization in the sustainable reports. A unit conversion was performed for each variable. In the independent variables, water use was measured in cubic meters per occupied room, carbon intensity was collected in kilograms of $CO_2e$ per occupied room, and RevPAR and NetRoom were presented in USD. The dependent variable, energy use, was defined in KWh per occupied room. Table 2 summarizes the variables' descriptive statistics.

**Table 2.** Variables' descriptive statistics.

| Variable | Valid n | Mean | Minimum | Maximum | Std. Deviation |
|---|---|---|---|---|---|
| NetRoom (USD) | 45 | 9821.5 | −47,272.7 | 238,718.9 | 36,788.6 |
| RevPAR (USD) | 45 | 141.1 | 28.6 | 808.6 | 129.0 |
| Energy (KWh/occupied room) | 45 | 101.0 | 9.3 | 636.2 | 121.3 |
| Water (m$^3$/occupied room) | 45 | 0.9 | 0.1 | 4.0 | 0.8 |
| Carbon (kgCO2e/occupied room) | 45 | 42.9 | 2.0 | 274.8 | 57.2 |

### 3.2. Multiple Regression Model

Initially, to examine the bivariate relationships between variables, the dispersion matrix is used, which is appropriate when multivariate techniques are employed. In the matrix, the scatterplots are displayed at the bottom of the matrix, while the distributions are displayed on the main diagonal, and the Pearson's correlations between the variables appear at the top [19]. Presenting the relationships provides a perspective on how the independent variables relate to each other and the dependent variable.

The linear connection among variables is estimated by the Pearson correlation coefficient (R), presented in Equation (1). Where R is the data's set goodness of fit, X and Y are random variables, n is the number of observations, and $\overline{X}$ and $\overline{Y}$ are arithmetic means of the observations [20]. This coefficient fluctuates between −1 and +1, presenting the correlation between the parameters. If the correlation between the variables is positive, values are between 0 and +1, if it is negative, values are between −1 and 0, and the correlation is null if the value is 0 [21].

$$R = \frac{\sum_{i=1}^{n} X_i Y_i - n\overline{X}\,\overline{Y}}{\sum_{i=1}^{n}\left(X_i^2 - n\overline{X}^2\right)\left(Y_i^2 - n\overline{Y}^2\right)} \tag{1}$$

Multiple linear regression was the method chosen to assess how the predictor variables (water use, carbon intensity, RevPAR, and NetRoom) are associated with the dependent variable (energy use). This statistical technique was employed because the data for all variables are metric and there is a clear definition between the dependent and independent variables. The decision process for multiple regression and the mathematical background, as in reference [19], were used in the development of the methodology. The multiple linear regression model is given by Equation (2).

$$Y = \beta_0 + \beta_1 X_1 + \beta_2 X_2 + \ldots + \beta_n X_n + \epsilon \tag{2}$$

where, $\beta_0$ is the intercept and $\epsilon$ is the error value. The expected value of the dependent parametric variable with error term is assumed to be zero, so the estimated multiple regression is obtained from Equation (3), where $b_0$, $b_1$, $b_2$ ... $b_n$ are estimates of $\beta_0$, $\beta_1$, $\beta_2$, ... $\beta_n$, and $\hat{Y}$ is the predicted value for the dependent variable [22].

$$\hat{Y} = b_0 + b_1 X_1 + b_2 X_2 + \ldots + b_n X_n \tag{3}$$

Each coefficient of the multiple regression model is interpreted as the estimated variation in the dependent variable corresponding to a unit difference in an independent variable when other variables are kept constant [23]. To represent the proportion of the dependent variable variation explained by the influence of independent variables, the multiple coefficient of determination ($R^2$) is used [19,24]. Equation (4) presents the formula for $R^2$ calculation.

$$R^2 = \frac{SSR}{SST} \tag{4}$$

SSR is regression sum of squares, and SST is the total sum of squares, which englobes the regression sum of squares plus the sum of squares due to error. For an acceptable model, the hypothesis is to obtain a high enough $R^2$. This coefficient increases as new independent variables are added to the model, but it is not used to verify the model acceptance when



more than one predictor variable is used. The solution for this is the use of the adjusted $R^2$, which weighs more heavily on the model when there are more predictors. It penalizes the model for having too many predictors [23]. Equation (5) shows the adjusted $R^2$ (where $p$ is the number of parameters and n is the number of observations).

$$\text{Adjusted } R^2 = 1 - (1 - R^2)\left(\frac{n - 1}{n - p}\right) \tag{5}$$

In developing the regression model, the goodness of fit model for the understanding of energy use and its key factors must be verified. The basic assumptions to be measured are the linearity of the measured phenomenon, the normality of the distribution of error terms, the constant variance of error terms, the independence of error terms, the outliers' presence, and the multicollinearity verification [19,23,25].

Linearity is verified by analyzing the residual plot against the variables present in the model. It is expected that a definite shape for the points on the graph cannot be found. The residual normality test can also be verified using a graphical method, comparing the accumulated frequency of standardized residuals with the normal curve [19,23]. The constant variance of residuals is homoscedasticity. This is a fundamental property that must be guaranteed to validate the analysis. The residuals (errors) plots are checked against the actual values of the sample and the values calculated by the regression equation. If the points are randomly distributed, there is homoscedasticity [19].

Independence of error terms is the absence of correlation between earlier or later values in the series (autocorrelation). This problem is also called serial correlation [19]. In the study case, the autocorrelation absence or not will be evaluated using the Durbin–Watson statistical test. The outliers' presence is checked by the standard residual values [25], which must be smaller than three standard deviations. As for multicollinearity, the ideal situation is to have several independent variables highly correlated with the dependent variable, but with less correlation between themselves. The simplest way to verify this assumption is by examining the correlation matrix of the variables. A measure used to express the degree of multicollinearity is the VIF (variance inflation factor), whose most common cut-off point is a value of 10 [19].

## 4. Results

### 4.1. Dispersion Matrix

The dispersion matrix facilitates the global understanding of the relationship between the variables of interest. Pearson's correlation coefficients are shown in the upper-right part of the matrix, where the highlighted coefficients correspond to those with statistical significance ($\alpha$-value = 0.05). The matrix's main diagonal, the distributions of the collected data, is noted, and on the left are the scatter diagrams that identify the relationships determined from the correlation coefficients. Figure 1 displays the the dispersion matrix.

Regarding the dependent variable, it was noticed that all the other independent variables showed a significant positive correlation. Special attention can be given to Carbon and RevPAR, which showed a very high correlation with Energy, but a high correlation was also observed between the two, which can generate multicollinearity issues in the multiple regression. The other two independent variables showed a significant correlation with Energy and with Carbon, but at a lower intensity. Therefore, by examining the dispersion matrix, it is understood that the regression analysis must be performed with all the variables proposed.

### 4.2. Multiple Regression—First Model

To analyze the model significance, an analysis of variance (ANOVA) is tested. The F-test is significant, with an F-value of 212.86 corresponding a $p$-value < 0.01. The coefficient of determination, $R^2$ = 0.9551, and the adjusted coefficient of determination, $R^2$ adjusted = 0.9506, are calculated. To predict the energy values in hotels, Equation (6) is

used, and all regression coefficients, including the intercept, showed statistical significance, with the *p*-value $< 0.05$.

$$\hat{Y}_{\text{Energy}} = -17.861 - 0.001\text{NetRoom} + 0.244\text{RevPAR} + 23.685\text{Water} + 1.620\text{Carbon} \quad (6)$$

From the values observed in the determination coefficients ($R^2$ and adjusted $R^2$), it can be identified that the model represents the sample and can represent the population at a 5% confidence interval level. This is an important outcome because it connects a representative number of hotels distributed worldwide. Table 3 reports the Model 1 regression summary and the analysis of variance.

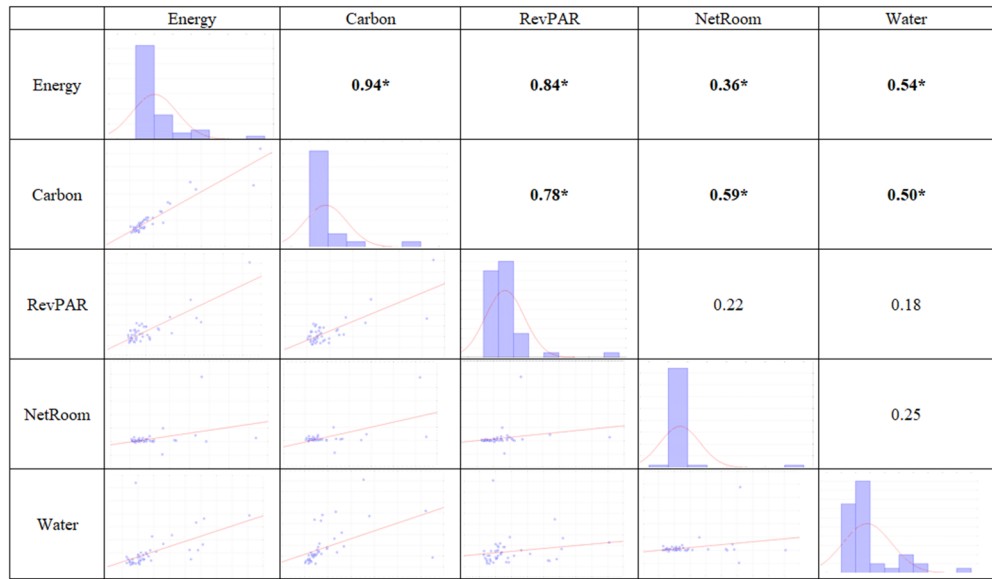

**Figure 1.** Dispersion matrix. (*) indicates *p*-value $< 0.05$.

The model adequacy is verified by the linearity assumptions of the measured phenomenon, the normal errors distribution, constant error variance, errors independence, and non-multicollinearity verification. In the linearity analysis, there was no form connecting the points, which leads to the fulfillment of the first adequacy assumption. The residual normality test was conducted by graphically comparing the residuals with the normal curve, which generated an approximation with a significance level of 95%. So, the linearity and normality assumptions for the model's adequacy are adequate. Figure 2 shows the linearity and normality tests.

**Table 3.** Model 1 regression summary and the analysis of variance.

| Panel A—Regression Summary | | | |
|---|---|---|---|
| **Variable** | **Coefficient** | *t*-**Statistic** | *p*-**Value** |
| Intercept | $-17.8615$ | $-2.2049$ | 0.033 |
| NetRoom | $-0.0006$ | $-3.8490$ | <0.001 |
| RevPAR | 0.2437 | 3.7953 | <0.001 |
| Water | 23.6850 | 3.7234 | <0.001 |
| Carbon | 1.6201 | 8.3709 | <0.001 |
| **Panel B—Analysis of Variance** | | | |
| F-Statistic | 212.86 | | |
| *p*-value | <0.001 | | |
| $R^2$ Adjusted | 0.9506 | | |

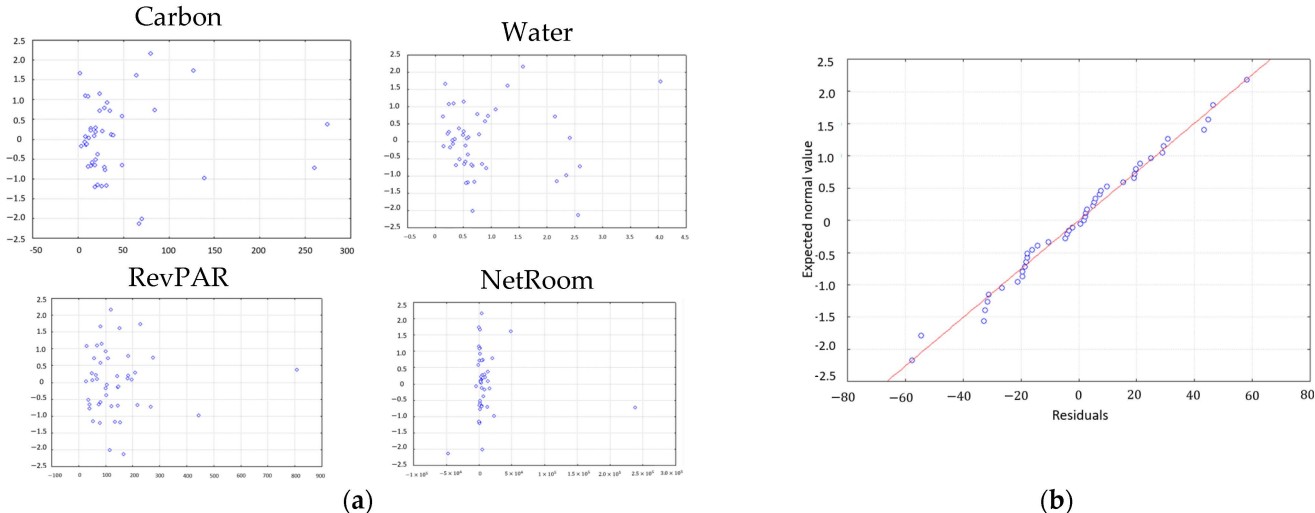

**Figure 2.** Linearity and normality tests for Model 1. (**a**) Phenomenon linearity; (**b**) residuals' normality.

To verify if the errors have a random behavior and are not related to the variables under study, the homoscedasticity analysis is conducted. The analysis was performed using a graph relating the residual against the predicted values, and its result is verified in Figure 3. The points are randomly distributed without a systematic behavior, indicating the presence of homoscedasticity. The correlation among residuals was estimated using the Durbin–Watson test, which generated a DW statistic of 2.05, corresponding to a serial correlation of −0.099. As this DW statistic value is within the critical value for the absence of autocorrelation (also verified with the small estimate of serial correlation), it is understood that there is no autocorrelation between the residuals of the model. The extremes standard residual values for this model were −2.14 and 2.16, indicating the outliers' absence.

The last assumption to be analyzed was related to multicollinearity. In this case, through the tolerance analysis and the generated VIF, values between 1.72 and 7.43 were checked. The model can be accepted following the cut-off point of 10. However, due to the small sample size, it was decided to develop a second model, excluding an independent variable with a high correlation with other independent variables. Thus, it was decided to exclude the independent variable RevPAR, which presented a high correlation with the variable Carbon and that had already shown significance in other studies involving the topic.

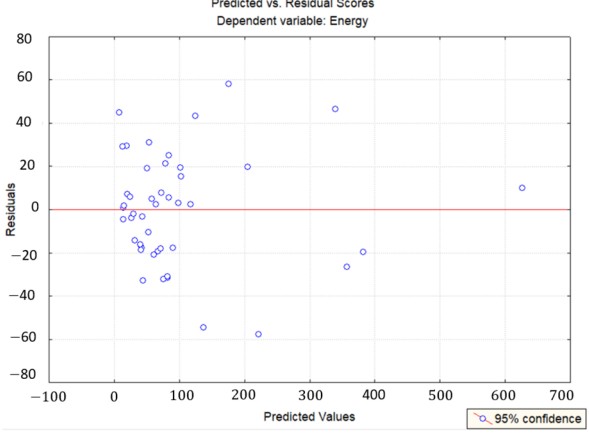

**Figure 3.** Homoscedasticity analysis for Model 1.

### 4.3. Multiple Regression—Second Model

When removing the RevPAR variable, it was noticed that the model intercept and the independent variable Water generated non-significant coefficients. Thus, the second

model was estimated with two independent variables (Carbon and NetRoom). As in the first model, the second was also significant, with an F-value of 295.10 and a *p*-value less than 0.01. The coefficient of determination is equal to 0.9336 and the adjusted coefficient of determination is 0.9304. The preliminary evaluation through the adjusted coefficient of determination suggests that the model maintains a high representation compared to the first model, even with two variables less. The regression formula is described in Equation (7) (the regression coefficients showed statistical significance, with a *p*-value < 0.05).

$$\hat{Y}_{Energy} = 9.668 - 0.001 NetRoom + 2.350 Carbon \tag{7}$$

It can be observed that the second model maintains the relationships previously verified between the independent variables and the dependent variable. The NetRoom had a relation intensity such as that verified in the first moment, with a negative relationship with the dependent variable. The variable Carbon, on the other hand, showed an increase in the intensity of the relation but maintained the same positive relationship verified in the first model. Table 4 presents the Model 2 regression summary and the analysis of variance.

**Table 4.** Model 2 regression summary and the analysis of variance.

| Panel A—Regression Summary | | | |
|---|---|---|---|
| **Variable** | **Coefficient** | ***t*-Statistic** | ***p*-Value** |
| Intercept | 9.6675 | 1.5907 | 0.119 |
| NetRoom | −0.0010 | −5.9775 | <0.001 |
| Carbon | 2.3502 | 22.5844 | <0.001 |
| Panel B—Analysis of Variance | | | |
| F-Statistic | 295.10 | | |
| *p*-value | <0.001 | | |
| $R^2$ Adjusted | 0.9304 | | |

The basic assumptions made to validate the first model were also performed to assess the suitability of the second model. Figure 4 presents the linearity, residual normality, and homoscedasticity tests for Model 2. It is noted that the model is adequate for the analysis assumptions of linear regression. The autocorrelation was verified through the value of the DW statistic of 2.03 and the serial correlation of −0.13, indicating the non-existence of serial correlation. The extremes standard residual values were −2.73 and 2.42. The VIF value in the second model was 1.52, a value considered acceptable for a multiple linear regression model.

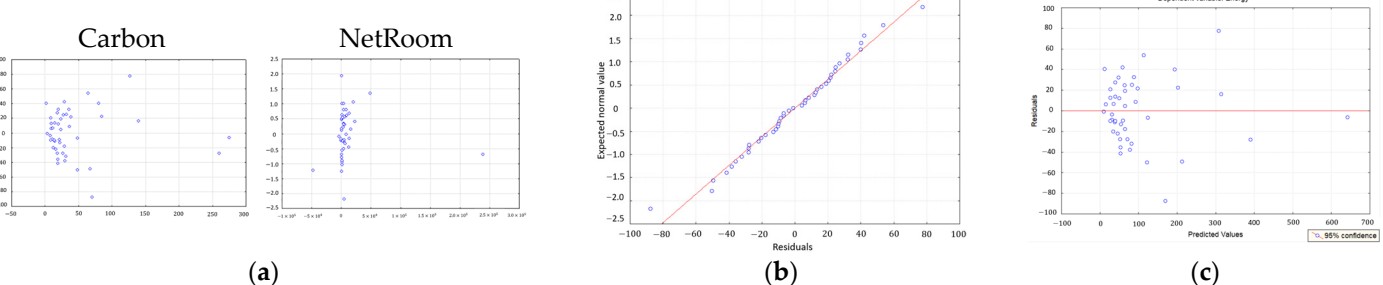

**Figure 4.** Linearity, residual normality, and homoscedasticity tests for Model 2. (**a**) Phenomenon linearity; (**b**) residuals' normality; (**c**) homoscedasticity.

## 5. Discussion

As verified in other works [4,13,14], it is concluded from the first regression model that higher revenues in hotels are linked to higher levels of energy use. This can be interpreted as the need for hotels to pass on the cost of energy to their customers, generating greater

RevPAR values in hotels with higher energy consumption. It can be inferred that the amount paid by hotel customers considers the energy consumption per occupied room. This result can be identified in the first regression model and may suffer some distortion because of its small sample and high VIF value. The confirmation in this model is relevant, as the previous literature supports this result.

Another relevant aspect verified by the first regression model was the relationship between water consumption and energy use. This relationship had not yet been investigated, at least in the studies found on the subject. Water consumption appeared together with energy as a dependent variable [12,13], without a study of the correlation between them. However, as expected, the greater the number of guests, the greater the revenue per room, and more elevated are the consumption of resources, including energy and water. The regression showed a significant positive relationship between water consumption and energy use, which demonstrates that when more water is consumed, more energy is used in hotels. This relationship was significant in the first model, while it was left out of the second one because the $p$-value of its coefficient was between 0.05 and 0.1. It can be concluded that it represents part of the population, with the exception that further studies can reassess this relationship.

Of the ten hotel chains with the lowest NetRoom values, two were among the ten largest consumers of energy, three were among the top ten water consumers, and three were in the emissions top ten. As for RevPAR, only one was among the ten largest energy consumers and two were among the ten largest water consumers. This supports the result that profitability is being affected in companies that consume more natural resources. The four hotel chains with the highest rates of carbon emissions launched approximately 41.5% of the total launches per occupied room in the sample, even though they had only 0.2% of the total rooms. The same networks consumed 37.7% of the energy per occupied room, and two of these were among the four largest consumers of water (29% of the sample).

Just as the use of natural resources through water represented a positive relationship with energy use, the emission of polluting gases also showed the same relationship. This relationship can be explained since more movement and use of facilities by hotels generated a greater consumption of energy sources and the emission of particles to the environment. How emission data is collected also considers the energy consumption of hotels, so it is expected that the relationship will be significantly positive. This was demonstrated in both models and was the strongest correlation found in the work, which shows that energy is highly correlated with carbon emissions.

This relationship further demonstrates the importance of campaigns to reduce the use of polluting energy sources and the replacement of outdated systems with modern ones, as seen in studies on the subject (references [26,27]). Solar energy is being widely used and developed for companies, both in business and in services. In addition, this type of energy generation is recommended for many applications related to hotels, such as heating water, and for lights, meals, maintenance, and so on [15]. Other minor aspects can be incorporated into the hotel system to reduce energy use. For example, air conditioning use can be optimized through real-time monitoring of weather conditions (a temperature controller triggered by the variation verified in the environment). Lights with presence sensors can be added in areas with little movement of people. Progressive discounts depending on the use of guests can be proposed, among other opportunities. The importance that the energy prediction model attaches to carbon emissions is memorable regarding the need to update the energy grid. While many hotel chains have implemented more sustainable forms of accommodation, it is still possible to notice those who are high consumers and who do not implement improvement projects in this aspect.

Updating hotels' energy grids will play a key role in their survival. This can be verified by analyzing the role of the net income variable in the regression models. Despite not having a high coefficient, its relationship with energy consumption was significantly negative in both models. This relationship shows that although revenues continue to increase according to energy use, profit does not increase, but decreases. This explains

the long-term importance of updating the energetic systems of companies and hotels. In addition to contributing to the more sustainable future of its operations, building positive marketing for its brands, and retaining a greater number of guests, it also has the potential for financial return.

The relationship verified through an indicator for net income per room had not been found in any other article on the subject. With the result verified in the study, the importance of including variables that may present significant relationships in the regression model was demonstrated. As the variables that are related to occupancy and hotel revenue proved to be significant, a test with a profitability variable was performed, showing a significant result. It is expected that further studies can be developed, both in hotels and other companies, verifying the adequacy of the independent variable proposed.

This research has some limitations. It is difficult to find hotel chains to participate with the disclosure of this type of data, which made sampling difficult. The study was conducted in a static moment; therefore, new studies can contribute with larger samples in larger periods to verify if the relationships are maintained. The GRI framework variables discussed in this article refer to financial and environmental aspects, and a new model incorporating other variables can be adopted, i.e., social variables. The relationship between financial, environmental, and social indicators can be evaluated to verify the importance of the triple bottom line concept in hotel chains' performances.

## 6. Conclusions

Energy use and some of its key factors were analyzed in this study that aimed for an understanding of its relationships. Data were collected on variables related to previous research on the theme and those present in the GRI framework, which provided a high level of reliability due to the third-party checks. Although the sample of hotel chains is not large (45 observations), it represents a considerable part of hotels in all regions of the world (approximately 54,000 properties). Due to the small sample, the study developed two multiple regression models to reduce the possible influence of multicollinearity.

From the analysis of linear regression models, concepts already identified in previous research can be reaffirmed, in addition to tracing new relationships involving energy use in hotels. It was seen that with higher energy use, the RevPAR indicator also increased but the NetRoom decreased. This demonstrates that hotels must pass on the price of energy consumption to their guests, but the return in profitability is not being generated. Other studies have highlighted the importance of updating the energy grid in hotel facilities, and our study presents an additional factor in identifying this need. By investing in updating their systems, hotel chains can contribute to the more sustainable future of their operations, build positive marketing for their brands, retain a greater number of guests, and generate a long-run financial return.

The environmental issue is also demonstrated in the work, since higher levels of energy use generate higher levels of emissions and water consumption. In this way, by constantly updating energy systems and monitoring energy use and water consumption, hotel chains can significantly reduce environmental degradation. This also helps to reinforce the positive points mentioned above. This work contributes to the scientific literature by confirming previously verified relationships and evidencing new ones between variables not yet explored. In addition, it is expected that the study will serve as a reference for policies to reduce energy use in hotels and for hotel owners and/or hotel chains to upgrade their systems.

**Author Contributions:** Conceptualization, R.S.A.; data curation, R.S.A.; investigation, R.S.A., A.M.S. and R.R.Z.; writing—original draft preparation, R.S.A.; writing—review and editing, A.M.S. and R.R.Z. All authors have read and agreed to the published version of the manuscript.

**Funding:** The study is supported by the Graduate Support Program (Programa de Apoio à Pós-Graduação—PROAP) of the Brazilian National Government.

**Institutional Review Board Statement:** Not applicable.

**Informed Consent Statement:** Not applicable.

**Data Availability Statement:** The data supporting can be found in open sustainable and annual reports of the most prominent hotel chains worldwide.

**Conflicts of Interest:** The authors declare no conflict of interest.

## Appendix A

**Table A1.** Data collected from sustainable and annual reports of hotel chains worldwide.

| Yi | Net Profit | RevPAR | Energy | Water | Carbon |
|---|---|---|---|---|---|
| H.1 | 22,878.30 | 444.28 | 331.34 | 2.35 | 139.44 |
| H.2 | 5774.09 | 276.00 | 225.35 | 0.95 | 84.73 |
| H.3 | 921.85 | 134.60 | 50.15 | 0.70 | 31.28 |
| H.4 | 4899.54 | 49.85 | 27.66 | 0.24 | 14.49 |
| H.5 | 6432.72 | 102.83 | 42.24 | 0.59 | 22.04 |
| H.6 | 713.02 | 154.00 | 43.79 | 0.58 | 26.60 |
| H.7 | 916.79 | 75.00 | 24.63 | 0.52 | 18.60 |
| H.8 | 1135.02 | 79.38 | 11.83 | 0.55 | 19.00 |
| H.9 | 3111.59 | 64.72 | 30.23 | 0.22 | 14.29 |
| H.10 | 1699.31 | 81.78 | 23.83 | 0.54 | 16.10 |
| H.11 | 3785.45 | 28.57 | 14.87 | 0.31 | 12.40 |
| H.12 | 909.50 | 41.64 | 72.22 | 0.84 | 48.40 |
| H.13 | 372.36 | 227.95 | 385.83 | 4.05 | 127.20 |
| H.14 | 1253.55 | 41.11 | 40.41 | 0.92 | 29.81 |
| H.15 | 3379.30 | 183.00 | 101.22 | 0.60 | 36.74 |
| H.16 | 9510.91 | 183.46 | 89.29 | 0.79 | 27.33 |
| H.17 | 15,864.20 | 146.00 | 22.45 | 0.15 | 9.31 |
| H.18 | 911.73 | 109.65 | 69.52 | 0.14 | 23.97 |
| H.19 | 303.39 | 85.41 | 84.30 | 0.51 | 23.81 |
| H.20 | 1828.67 | 67.77 | 48.38 | 0.33 | 8.42 |
| H.21 | 1504.88 | 80.90 | 53.18 | 0.18 | 1.96 |
| H.22 | 5730.57 | 144.51 | 62.65 | 0.50 | 19.83 |
| H.23 | 20,119.99 | 183.60 | 100.64 | 0.75 | 29.06 |
| H.24 | 4160.77 | 120.09 | 234.29 | 1.58 | 80.20 |
| H.25 | 238,718.90 | 266.70 | 362.50 | 2.60 | 260.27 |
| H.26 | 1863.48 | 101.61 | 108.66 | 1.08 | 32.30 |
| H.27 | −47,272.73 | 166.50 | 163.99 | 2.56 | 67.22 |
| H.28 | 471.24 | 30.71 | 42.58 | 0.24 | 11.41 |
| H.29 | 8238.21 | 210.65 | 80.46 | 0.51 | 19.89 |
| H.30 | 8524.15 | 101.85 | 9.29 | 0.26 | 3.69 |
| H.31 | 12,614.36 | 122.98 | 47.83 | 0.67 | 29.00 |
| H.32 | 13,247.55 | 808.62 | 636.18 | 0.43 | 274.83 |
| H.33 | 2762.43 | 69.85 | 120.00 | 2.41 | 38.90 |

**Table A1.** *Cont.*

| H.34 | 3955.69 | 217.42 | 53.17 | 0.65 | 14.41 |
| H.35 | −3979.84 | 104.27 | 28.31 | 0.32 | 7.85 |
| H.36 | 4311.40 | 148.00 | 40.37 | 0.54 | 9.47 |
| H.37 | 1614.62 | 37.41 | 17.71 | 0.44 | 19.28 |
| H.38 | 49,037.36 | 153.00 | 167.76 | 1.30 | 64.57 |
| H.39 | 4398.52 | 58.46 | 121.44 | 2.14 | 35.60 |
| H.40 | 121.75 | 54.43 | 51.11 | 2.19 | 21.70 |
| H.41 | 4564.21 | 145.34 | 22.80 | 0.37 | 11.14 |
| H.42 | −587.73 | 82.26 | 118.16 | 0.89 | 48.80 |
| H.43 | 2664.17 | 52.16 | 17.29 | 0.35 | 8.57 |
| H.44 | 13,458.34 | 196.08 | 66.95 | 0.57 | 18.31 |
| H.45 | 5124.64 | 115.84 | 82.27 | 0.67 | 70.23 |

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
