# Peer review of "Energy Use and Its Key Factors in Hotel Chains"

_sustainability, doi:10.3390/su14148239_

Round 1

Reviewer 1 Report

Paper is focused on the Energy consumption in hotels. While they consume a lot of Energy, from this reason the topic is important, up-to-date and worth of investigating. Paper lies in the scope of Energies journal.

Paper is well written. Methods seems to be appropriate. Limitations of the work are listed. Structure of the paper is good.

Paper brings limited contribution to the fired of energy use and does not bring any practical solution how to save the Energy or make processes more effective. Nevertheless, paper describes interesting approach to the field of making hotels more energy efficient. From this reason I recommend the paper for publication after some changes.

 1) Abstract. Please put more quantitative findings of the work and briefly describe the calculation methods used.

 2) The last paragraph with limitation I suggest to put as a last paragraph of discussion.

 3) More quantitative conclusions should be presented. Please prepare additional comparisons, some percentage differences. There is a lack of quantitative conclusions which should contain main findings from the paper and highlight the new and contribution of your work to the field. Please compare your results with other results from the literature.

 4) What about practical application? Does the observed correlations show how to make hotles more energy-efficient, what systems to use, in what places and with what devices / systems / behavior changes / control improvements to introduce changes in order to achieve higher energy efficiency at the lowest financial energy and environmental costs?

Author Response

We would like to thank you for your review. It brought important and more detailed and revised information about what was done and significantly increased the quality of the paper. Thank you so much for your time and dedication.

Reviewer 2 Report

Dear authors

Thank you so much for submitting your exciting manuscript to sustainability. I have carefully read your paper and I hope the following suggestions improve the quality of your paper.

Abstract:

- Please mention the complete words of RevPAR

Introduction:

- The gap and contributions of the current study should be highlighted! In the last paragraph, the authors mentioned "This research intends to provide a better understanding of energy use and its key" How and why? These are required!

The methodology section is very good

The Discussion Section is also excellent

Conclusion:

- Theoretical and practical implications should be enhanced and detailed

Best Wishes

Author Response

(The authors gave the same response as above.)

Round 2

Reviewer 1 Report

Revised version of the paper is improved.
I have no farther suggestions for Authors.

Paper should be considered for publication, becasue:
+ brings quantitative analysis of the sesitivity analysis of choosen pramters on the energy use in hotels
+ topic lies in the scope of the journal
+ paper is well written and well organized
+ paper seems to be interesting for Readers
+ paper presents an universal method that can be used in different branches for simmilar analysis
+ conclusions can be used for farther investigations by Authors or other reserachers